# Telomere-to-telomere genome sequence of the model mould pathogen *Aspergillus fumigatus*

Paul Bowyer [1], Andrew Currin [2], Daniela Delneri [2] & Marcin G. Fraczek [2]

The pathogenic fungus *Aspergillus fumigatus* is a major etiological agent of fungal invasive and chronic diseases affecting tens of millions of individuals worldwide. Draft genome sequences of two clinical isolates (Af293 and A1163) are commonly used as reference genomes for analyses of clinical and environmental strains. However, the reference sequences lack coverage of centromeres, an accurate sequence for ribosomal repeats, and a comprehensive annotation of chromosomal rearrangements such as translocations and inversions. Here, we used PacBio Single Molecule Real-Time (SMRT), Oxford Nanopore and Illumina HiSeq sequencing for de novo genome assembly and polishing of two laboratory reference strains of *A. fumigatus*, CEA10 (parental isolate of A1163) and its descendant A1160. We generated full length chromosome assemblies and a comprehensive telomere-to-telomere coverage for CEA10 and near complete assembly of A1160 including ribosomal repeats and the sequences of centromeres, which we discovered to be composed of long transposon elements. We envision these high-quality reference genomes will become fundamental resources to study *A. fumigatus* biology, pathogenicity and virulence, and to discover more effective treatments against diseases caused by this fungus.

*Aspergillus fumigatus* causes over 11 million allergic and over 3 million chronic and invasive lung infections annually, representing a significant complication of profound immunosuppression, chronic obstructive pulmonary disease (COPD), severe viral respiratory infections (such as influenza or Covid-19) and many other pre-existing conditions[1–4]. Mortality rates with effective treatment for invasive disease remain ~50%[5] and >80% for individuals infected with drug resistant isolates[6]. *A. fumigatus* is arguably the model human mould pathogen, with extensive research being carried out to understand its pathogenicity. The availability of *A. fumigatus* genome sequence has underpinned many of the rapid advances in our understanding of this organism in recent years.

The first *A. fumigatus* genome sequence was published in 2005[7] for a clinical isolate Af293, followed by the A1163 strain published in 2008[8]. These two reference genome sequences have been crucial to study the biology and pathogenicity of this fungus. However, due to the technological capabilities at the time, the original reference sequences are not complete with absent sequences (deletions) or gaps filled with unknown nucleotides (NNN). The Af293 assembly benefited from extensive manual annotation and addition assembly experiments such as optical mapping, whereas A1163 remains a series of unlinked contigs[8]. Moreover, these sequences lack coverage of centromeres, an accurate sequence for the ribosomal repeats, and a comprehensive annotation of chromosomal rearrangements such as translocations and inversions. A1163 or strains derived from its parental isolate

[1]Manchester Fungal Infection Group, Division of Evolution, Infection and Genomics, School of Biological Sciences, Faculty of Biology, Medicine and Health, University of Manchester, Manchester, UK. [2]Manchester Institute of Biotechnology, Faculty of Biology, Medicine and Health, The University of Manchester, Manchester, UK. e-mail: paul.bowyer@manchester.ac.uk; d.delneri@manchester.ac.uk; fraczekmg@gmail.com

CEA10[9,10] have become standard in laboratory experiments because of their robust pathogenicity and growth. For example, the CEA10 descendant isolate A1160, recently renamed to MFIG001[10], is a standard laboratory isolate first mutated from CEA10 to uridine auxotrophy (*pyrG*⁻) form CEA17[11] and subsequently used to construct the *pyrG*⁺ *ku80* knockout strain A1160[12]. This strain is currently being used as a host strain for a whole genome knockout project[13,14] and forms the basis of many virulence, transcriptomics and other experiments[15]. Therefore, there is an urgent requirement to revise the original genome sequences and provide comprehensive genome assemblies of the most exploited *A. fumigatus* strains A1160 and CEA10 using the current long-read next generation sequencing technology.

Recent advances in the long read next generation sequencing technologies, such as Pacific Biosciences (PacBio) and Oxford Nanopore, allow longer reads and more accurate assembly of genomic sequences. They have been used to provide complete and accurate genome assemblies of a wide range of organisms, including human, plants and animals as well as fungal pathogens such as *Magnaporthe oryzae* and *Aspergillus awamori*[16–18]. Due to the pathogenic nature of *A. fumigatus*, with large numbers of patients suffering from aspergillosis worldwide as well as increasing numbers of fungal studies, there is an urgent requirement for the assembly of high-quality reference genomes of commonly used *A. fumigatus* isolates.

In the present work, by using both PacBio and Nanopore technologies, we carry out genome sequencing and assembly of two *A. fumigatus* strains, CEA10 and A1160. The obtained CEA10 data are also subsequently polished using previously generated in-house Illumina HiSeq sequences. By combining these three genome sequencing technologies, we present the complete high quality de novo telomere-to-telomere genome sequence of CEA10 and near complete assembly of A1160 revealing centromere structure, ribosomal repeat sequence and chromosomal organisation. As previously predicted[8], CEA10 shows chromosomal rearrangements when compared to Af293. Moreover, there is evidence of a small

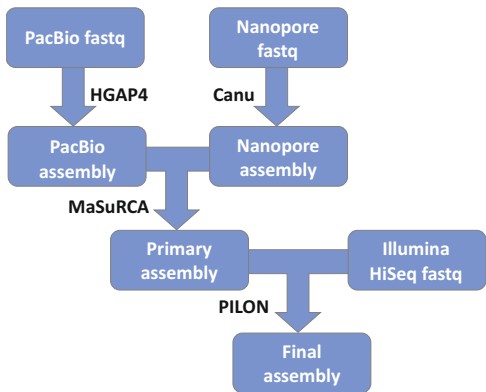

**Fig. 1 | Pipeline for assembly of Aspergillus fumigatus CEA10 and A1160 genomes.** PacBio and Oxford Nanopore reads were assembled using PacBio HGAP4 and Canu[19,20], respectively. The assemblies were not full length at this point. The 5' end of chromosome 6 and the rRNA repeat region caused breaks in the sequence for PacBio assemblies of both A1160 and CEA10. Oxford Nanopore assemblies included these regions but did not cross the centromere for chromosome 4 for both strains. PacBio and Oxford Nanopore assemblies were then combined using MaSuRCA[21] to give primary assemblies for both CEA10 and A1160. The primary CEA10 assembly consisted of 8 chromosomes with telomere sequence for both chromosome ends and no gaps. The primary A1160 sequence consisted of 6 complete chromosome regions using the CEA10 sequence as comparator lacked Oxford Nanopore coverage of the 5' end of chromosome 6 or centromeres of chromosomes 1, 2, and 5. Finally, the CEA10 assembly was polished using Illumina HiSeq 2500 paired end 2 × 150 reads from two CEA10 libraries with predicted 73 and 81 fold coverage of the genome. PILON was used for rounds of assembly until no further improvements in the sequence were observed (4 rounds in total).

number of mutations, potentially affecting gene function that have accrued in the last ~30 years since isolation of CEA10, and the creation of A1160 in the laboratory. The sequences obtained and analysed in this study are now publicly available for the scientific community and will greatly contribute to the future research on this fungus.

## Results and discussion

### Sequencing and de novo genome assembly

The complete genome sequence of two *A. fumigatus* laboratory reference strains, A1160 and CEA10 was carried out using the long read de novo PacBio and Oxford Nanopore next generation sequencing technologies. Additionally, previously generated in-house Illumina HiSeq data for CEA10 was used to further validate the final sequence of this strain. The workflow used to assemble the genomes are shown in Fig. 1. The data acquired allowed us to greatly improve the quality of the genome assembly compared to the original reference sequences of Af293 and A1163[7,8] and expand the genomic resources for this pathogen. Specifically, missing gaps were filled and additional genomic information on ribosomal repeats and centromere composition was added. Interestingly, we found that the centromeres of *A. fumigatus* encompass long stretches of DNA and are enriched with transposons. Moreover, the comparative analysis of A1160 and CEA10 vs Af293 revealed several chromosomal rearrangements, the largest of which is between chromosomes 1 and 6.

The PacBio and Oxford Nanopore sequencing generated sufficient data to allow high quality genome assemblies of the expected >29 Mb size[7]. Both strains were assembled in 10 contigs with 233x and 183x coverage for A1160 and CEA10 genomes, respectively, using the PacBio assembly algorithms and Canu[19,20] (Supplementary Data 1). GC content for both strains was ~49.5%. For the Oxford Nanopore sequencing, the same genomic DNA was used unsheared which provided longer raw data reads with N50 of 20 kB. The mean coverage for the Oxford Nanopore assembly using Canu 1.9 was 39x for both strains with 23 and 19 contigs for A1160 and CEA10, respectively. PacBio and Oxford Nanopore sequences were subsequently combined using MaSuRCA[21] to give primary assemblies for both CEA10 and A1160. Previously obtained Illumina HiSeq sequences were also used to validate CEA10 assembly with mean coverage 73x and 81x (Supplementary Data 1).

Our data show that the genomes of A1160 and CEA10 are almost identical in sequence besides a small number of single nucleotide polymorphism (SNP) variations (96) in several genes (Supplementary Data 2). The most evident changes in the SNPs are observed on chromosome 8, for which we also observed several insertions and deletions (INDELs) of nucleotides, leading to frame shift. There is a total of 34 INDELs between these 2 strains. For the strain A1160 the telomere on chromosome 6 could not be completely assembled due to chromosomal rearrangements.

Ribosomal sequence was extracted from the raw data using grep to capture reads known to contain *A. fumigatus* ribosomal sequences. For Oxford Nanopore data, assembled repeat regions were obtained as assembled contigs. The core assembly indicated only a single 28 S repeat and this is likely due to mis-assembly of the repeat units. As the number of repeats is not clearly distinguishable, the 28 S segment was left as a marker for the region on chromosome 4.

The mitochondrial sequences of both strains were also analysed, and we found that our assembled data are consistent with previously published sequences for A1160 and Af293[22].

### The new genome assembly unravels previously undetected gene sequences and chromosomal rearrangements

The original sequence of Af293 was created in 2005 using the whole genome random sequencing method[7]. Although, it still provides crucial sequencing data, it does not include centromeres or chromosomal

**Table 1 | Assembly statistics for assembled chromosomes**

| Chromosome | CEA10 | | | A1160 | | | Af293 | |
|---|---|---|---|---|---|---|---|---|
| | Size (bp) | Gene count | | Size (bp) | Gene count | | Size (bp) | Gene count |
| | | GMEP/Braker | FDB | | GMEP/Braker | FDB | | FDB |
| 1 | *4910137* | 1671 | 1671 | 4910149 | 1684 | 1671 | 4918979 | 1612 |
| 2 | *4786990* | 1607 | 1635 | 4786411 | 1617 | 1634 | 4844472 | 1624 |
| 3 | *4270437* | 1477 | 1452 | 4271837 | 1479 | 1453 | 4079167 | 1362 |
| 4 | *3819196* | 1285 | 1289 | 3812114 | 1288 | 1290 | 3923705 | 1224 |
| 5 | *3799902* | 1330 | 1345 | 3799872 | 1339 | 1345 | 3948441 | 1351 |
| 6 | *3900669* | 1330 | 1281 | 3848548 | 1332 | 1282 | 3778736 | 1227 |
| 7 | *1884953* | 614 | 608 | 1884753 | 617 | 608 | 2058334 | 621 |
| 8 | *1949143* | 642 | 636 | 1944132 | 641 | 631 | 1833124 | 609 |
| Total | *29321427* | 9956 | 9917 | 29257816 | 9914 | 9997 | 29384958 | 9630 |

Chromosome sizes in base pairs (bp). Genes were predicted for each chromosome either by using a GenemarkEP + /Braker pipeline (GMEP/Braker) or via transfer of annotations from the Af293 FungiDB database using Exonerate (FDB). New CEA10 chromosome and genome sizes after polishing with PILON are shown in italic.

rearrangements. In Table 1 we summarise the predicted sizes of chromosomes and genes from our PacBio analysis for A1160 and CEA10 and compare them to the sizes present in the database for Af293. Two different pipelines were used for gene annotation in this analysis revealing no major differences in chromosome sizes or gene complement between the previously generated reference sequences and our newly assembled genomes. As previously shown[7], the genome of *A. fumigatus* Af293 is arranged in 8 chromosomes of a total of approximately 29.2 Mb and our CEA10 sequence is comparable in size and chromosome number.

Protein coding gene transcripts, and transposons were annotated based on our de novo analysis and the data from FungiDB (Fig. 2). When determining centromere localisation, we observed that transposable elements, besides being scattered throughout the whole genome as predicted were also localised in the centromeres of all 8 chromosomes, forming the majority of centromeric sequences. Although, it was previously predicted that centromeres of filamentous fungi may be composed of transposons[23], our study is the first to confirm that the centromeres of *A. fumigatus* chromosomes are enriched with transposable elements. An example of a detailed chromosomal annotation is presented in Fig. 3.

Our sequencing data also confirmed the localisation of the native *ku80* gene deletion in CEA10[9] as well as the replacement of this gene in A1160 with *pyrG*[+] on chromosome 2[12] (Fig. 4). This observation and the relatively low number of variants between CEA10 and A1160 is remarkable given the long time period of laboratory manipulation for A1160; at least one UV mutagenesis and two transformations have been performed on this isolate in this period and the strain has been through almost 30 years of culture and storage.

The comparison between the genomes of the reference strain Af293 and sequenced CEA10/A1160 revealed a number of chromosomal rearrangements (presented in Fig. 5a, b as Mauve and SyMap plots[24,25]). The largest rearrangements are between the ends of chromosomes 1 and 6 (a situation previously suggested in the original A1163 sequencing[8]). Chromosomal rearrangements and chromosomal breakpoint usage have been proposed to play a significant role in evolution that lead to environmental adaptation and these events have been previously observed in filamentous fungi[26–28]. As both A1160 and CEA10 strains have been widely used for > 20 years, it is expected that they might have accrued mutations and chromosomal rearrangements.

## Conservation of translocation breakpoints in other genomes in the species

Translocation breakpoints detected in the Af293:CEA10 comparison were mapped and flanking sequence were determined for both

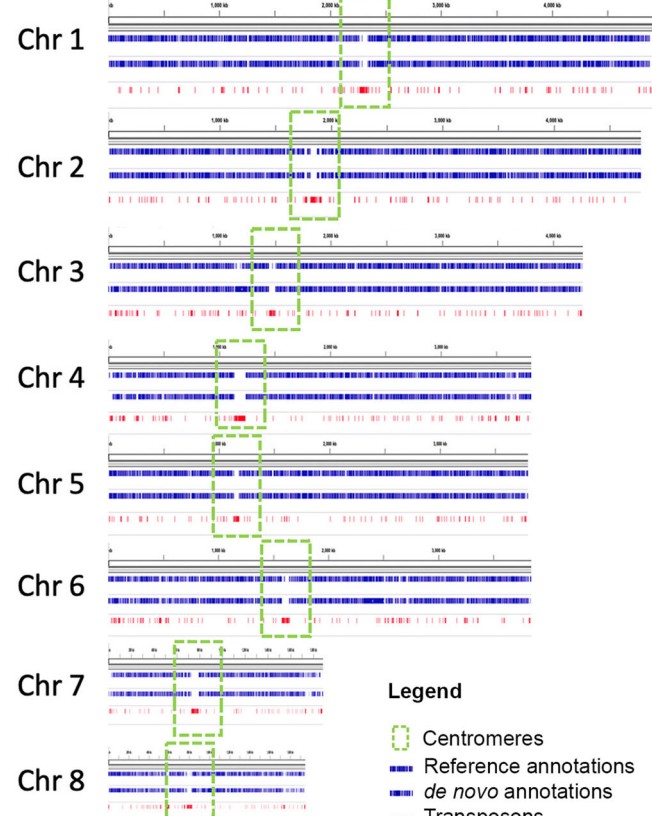

**Fig. 2 | Annotation of A1160 genome assembled in eight chromosomes.** Genomes are annotated in a cursory manner using Exonerate mapping of transcripts and proteins from Fungi DB and de novo gene prediction using Genemark EP + and Braker1 and Braker2. Transposons are mapped using Exonerate (Transposons). Genes are shown as blue bars and transposons are shown as red bars. Putative centromeres, enriched with transposons, are indicated by green rectangular boxes. Random PCR RNAseq data from NCBI was merged and mapped to genomes using HiSat2. Chr – chromosome.

species. Only breakpoints from translocations >100kB were included (Fig. 6a). Further translocations were identified but one or both flanking sequences contained repetitive DNA which hindered the comparative analysis. The mapped translocation events are complex and cannot be explained through direct Af293:CEA10 translocation. This is unsurprising given that the isolates have no known relationship and we suggest that these represent two instances of a complex

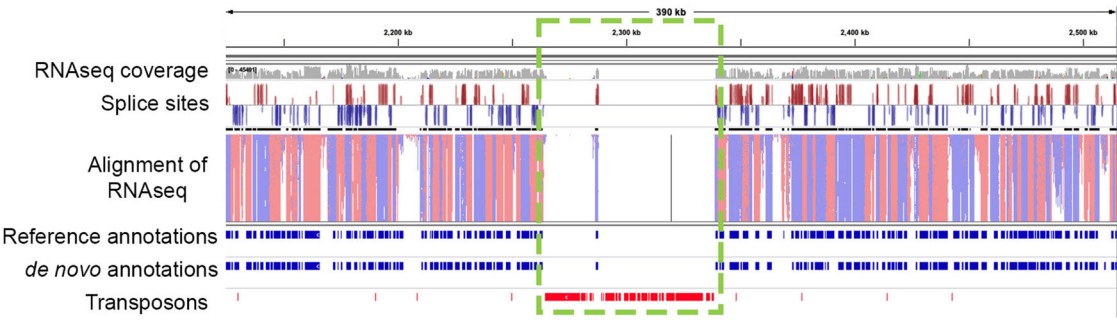

**Fig. 3 | Annotation of CEA10 genome showing a detail of chromosome 1 as an example, with a putative centromere.** Genomes are annotated in a cursory manner using Exonerate mapping of transcripts and proteins from (Fungi DB) and de novo gene prediction using Genemark EP + and Braker1 and Braker2. Reference and de novo gene annotations are shown in blue bars. The whole RNAseq coverage is shown in grey. Transposons are mapped using Exonerate (Transposons) with positions shown in red and the putative centromere enriched with transposons is shown in the green rectangle. Random PCR RNAseq data from NCBI was merged and mapped to genomes using HiSat2.

translocation landscape. Translocation regions consisting of 200 bp upstream and downstream of the breakpoint were compared to all 261 *A. fumigatus* genome assemblies available in NCBI (Supplementary Data 4). Several types of breakpoints can be observed as shown in Fig. 6b. Firstly, intact breakpoints, where both flanking regions or the query and the breakpoint are conserved (e.g. breakpoint 1 in Fig. 6b), show that most *A. fumigatus* isolates contain the breakpoint 1 structure from Af293 and the breakpoint 12 structure from CEA10. Numerous instances where no breakpoints or flanking regions are found can also be observed. Many genomes contain both flanking regions of the breakpoint but with the flanks matching different regions in the target genome (e.g. breakpoint 11) suggesting that translocation at the breakpoint has occurred but to different regions of the genome than observed for Af293:CEA10. Finally, many genomes contain one flank of the breakpoint or the other but not both, again suggesting different translocations from the same breakpoint but with loss of one flanking sequence. All translocation breakpoint flanking sequences from Af293 and CEA10 are listed in Supplementary Data 5.

The data shown in Fig. 6 suggests that the translocation breakpoints seen in the Af293:CEA10 comparison are common across *A. fumigatus* isolates. Moreover, it suggests breakpoint reuse in their evolutionary history. Whether common breakpoints in independent lineages are due to chromosomal site fragility or are a signature of a potential adaptive karyotypes remain to be investigated.

The availability of comprehensive genome sequence of *A. fumigatus* strains is crucial to understand the biology, pathogenicity and virulence of this fungus. Moreover, quality genome sequences are proving to be a powerful method for discovering mechanisms of drug resistance and may lead to more efficient patient treatment and their recovery. Here, we provide the comprehensive, telomere to telomere genome sequence of a widely used isolate of *A. fumigatus*, CEA10, and a near complete assembly of its descendant, A1160. This assembly has enabled us to fill in the gaps in the sequences of the original reference strains, Af293 and A1163. Our data shows significant improvement in sequence quality and organisation of chromosomes, revealing centromere structures, ribosomal repeats and breakpoints. The assembled sequences in this study should prove valuable to the scientific communities that lead research into better treatment and diagnostics of fungal diseases.

## Methods
### Strains and genomic DNA preparation
Two strains of *A. fumigatus*, CEA10 and A1160[10,12] were used in this study (available from The Fungal Genetics Stock Center - https://www.fgsc.net/). Fungal spores were used to extract high quality genomic DNA following a previously described CTAB method[12] with few

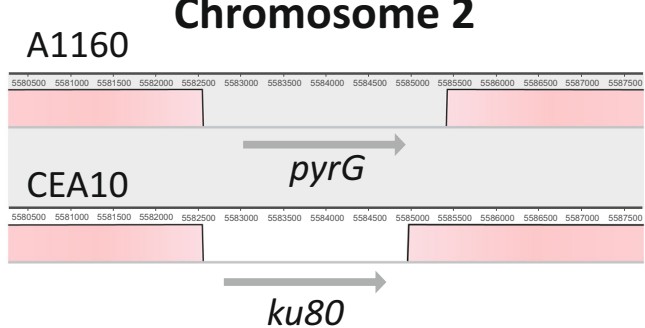

**Fig. 4 | Detail of chromosome 2 map showing gene replacement of ku80 in the A1160 strain with reference to the parental CEA10 isolate.** The native CEA10 *ku80* gene was replaced with a *pyrG* marker during construction of A1160[10]. Replacement sequences match precisely to the construct used to make the gene replacement and there are no variants in the sequence in this region.

modifications that greatly improved the quality and purity of extracted DNA. Briefly, both isolates were grown on SAB agar media in tissue culture flasks to minimise cross-contamination and spores were harvested in PBS/Tween20 and transferred to 2 ml screw top tubes containing 425−600 mm washed glass beads (filled to the 300 μL mark; ~50 mg) (Merck). Spores were centrifuged at max speed for 2 min using a benchtop centrifuge and the supernatant was removed. 1 mL of CTAB extraction buffer (2% CTAB, 100 mM Tris, 1.4 M NaCl and 10 mM EDTA, pH 8.0) was added and the tubes and they were vortexed at max speed for 10 min. Subsequently, the tubes were incubated for 10 min at 65 °C. Then, the above vortexing and heating process was repeated, and tubes were centrifuged at max speed for 2 min. The supernatant was transferred to new 2 ml tubes and an equal volume of chloroform was added. Tubes were mixed by inversion and centrifuged for 3 min at max speed. The aqueous phase was transferred to new 1.5 mL tubes and DNA was precipitated by addition of 0.6 volumes of isopropyl alcohol. Following centrifugation for 2 min at max speed, the supernatant was decanted, and the pellet was washed with 0.5 mL absolute ethanol. The pellet was briefly air-dried and resuspended in 200 μL of dH$_2$O. Subsequently, 2 μl of 100 mg/mL RNase A (Qiagen) was added and the tubes were incubated at 37 °C for 15 min. Then, 1 mL of buffer PB or PM (Qiagen), containing a high concentration of guanidine hydrochloride and isopropanol was added and mixed by pipetting. The solution was transferred onto silica based blue columns (NBS biologicals) and centrifuged for 30 sec at max speed. Then, 700 μL of buffer PE (Qiagen) was added onto the column and centrifuged as above followed by additional spinning for 1 min at max speed. The DNA

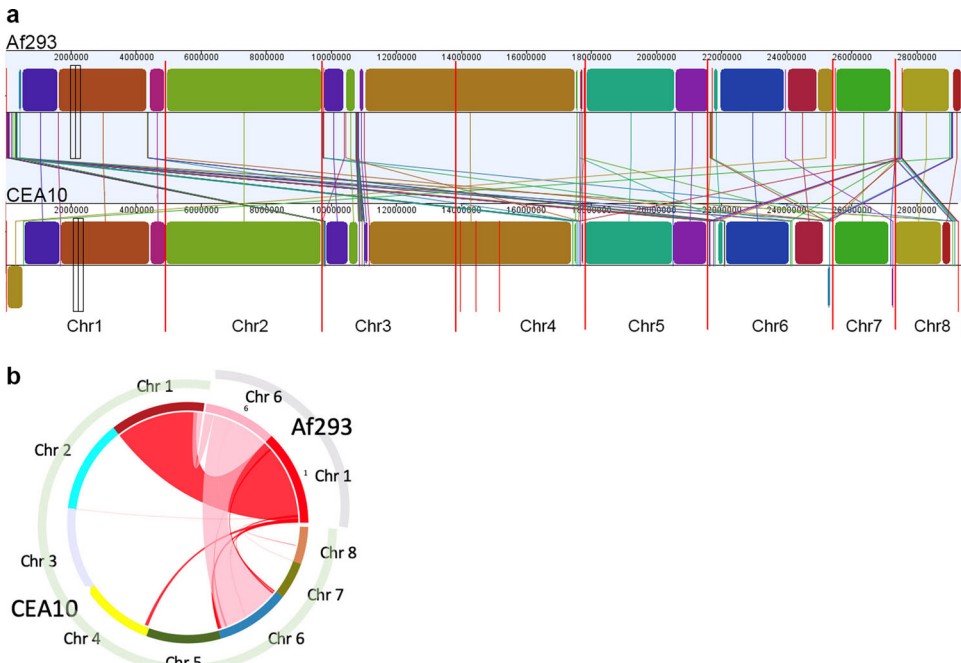

**Fig. 5 | Chromosomal rearrangements observed between Af293 and CEA10.** **a** Mauve plot of synteny[24] between Af293 and CEA10. Chromosomes are marked along the bottom of the panel. Syntenic blocks are shown as different coloured boxes for both strains with identical colours indicating synteny. Connecting lines are shown to indicate chromosome rearrangement. Note that the threshold for similarity in panel a is set to an LCB weight of 250000 which may include short homologous regions such as transposons **b** Circular SyMAP plot[25] showing large chromosome rearrangements between chromosome 1 and 6 in both Af293 and CEA10 strains. Chr – chromosome.

was eluted in 100 µL of dH₂O and the quality of the DNA was assessed on a 1% agarose gel, as well as using a nanodrop (Thermofisher Scientific) and a Qubit 4 Fluorometer (Thermofisher Scientific) to be within quality specification range required by the PacBio and Oxford Nanopore protocols.

**Library preparation for long read next generation sequencing**
For PacBio sequencing, genomic DNA was adjusted to 10 ng/µL in 150 µL volume and sheared to approximately 10 kb fragments using g-TUBES (Covaris) following the manufacturers' instructions. The size of fragments and quality of the DNA was verified using a Fragment Analyzer (Agilent) and the DNF-930 protocol. Samples were prepared for sequencing following the Express Template Prep Kit 2.0 protocol, with multiplexing using the Barcoded Overhang Adapter kit 8 A (both Pacific Biosciences). DNA libraries were sequenced using the SMRT Cell 1 M chips on the Pacific Biosciences Sequel system with 10 h data acquisition time.

For Oxford nanopore sequencing, 1 µg of the same DNA samples (not sheared) were prepared for sequencing using the SQK-LSK109 Ligation sequencing kit and Flongle sequencing expansion kit, following the manufacturer's instructions. Each strain was sequenced using a MinION Flongle flow cell with 24 h data acquisition time.

Previously generated HiSeq 2500 Illumina paired end reads of CEA10 were used here to validate and polish the final sequence.

**Genome assembly**
Pipeline for assembly of *Aspergillus fumigatus* CEA10 and A1160 genomes is summarised in Fig. 1. Demultiplexing and de novo assembly was performed using the Pacific Biosciences algorithms within the SMRT Link 8.0 software package. For de novo assembly the Hierarchical Genome Assembly Process (HGAP4) was used, with 30x seed coverage specified for each assembly with specified genome length of 29 Mb (all other parameters were unchanged). Assembly polishing and resequencing was performed using the Resequencing algorithm in SMRT Link 8.0.

For Oxford Nanopore data, base calling was performed using Guppy (Oxford Nanopore) and de novo assembly was performed using Canu 1.9[19,20], with specified genome length of 29 Mb. PacBio and Oxford Nanopore assemblies were then combined using MaSuRCA 4.0.9[21] to give primary assemblies for both CEA10 and A1160.

For CEA10, PacBio and Oxford Nanopore sequence assemblies were then polished using 3 rounds of PILON 1.24[29] with 2 paired end Illumina 2 × 150 fastq libraries (Fig. 1) to give the final CEA10 sequence.

**Annotation**
Genomes were subjected to a cursory annotation using a Genemark EP + pipeline[30] guided by Prothint 2.5.0 using orthodb version 10.1 as previously described[31]. Additionally, Augustus 3, BRAKER1 and 2 annotations were performed according to the software defaults for *A. fumigatus* and fungi, respectively[32]. Finally, an existing curated annotation for A1163 was mapped to the A1160 and CEA10 genomes using Exonerate 2.4.0[33]. Transposon sequences were collected for *A. fumigatus* from NCBI searches and further mapped onto the genome sequences using Exonerate. Transcript data from NCBI SRA (Supplementary Data 3) archive was used to guide annotation and to generate a list of potential transcribed regions which were then tested for the presence of ORFs, ORFs matching known proteins in the UniRef90 dataset or ORFs with matching PFAM domains using TransDecoder (https://github.com/TransDecoder/TransDecoder).

**Chromosome rearrangements and breakpoints between species**
Translocation breakpoints identified by the comparison of CEA10 and Af293 were mapped to other published *A. fumigatus* genome sequences using BLASTN. A number of potential translocation breakpoints are apparent in the comparative analysis of these two strains (Fig. 6). To further analyse occurrence of these breakpoints in the *A. fumigatus* community we compared breakpoint adjacent sequences with the 261 *A. fumigatus* genome assemblies present in the NCBI assembly database (https://www.ncbi.nlm.nih.gov/assembly) (Listed in Supplementary Data 4). Breakpoints were chosen to represent

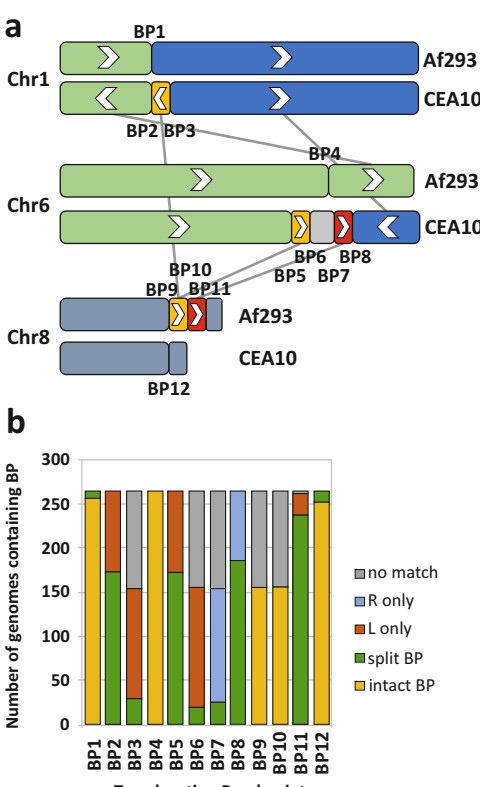

**Fig. 6 | Conservation of translocation breakpoints in other A. fumigatus genomes.** Translocation breakpoints found in the CEA10:Af293 comparison were mapped to other published *A. fumigatus* genome sequences. **a** Breakpoints for each translocation identified from the genome comparison are labelled as BP. **b** Presence of translocation breakpoints in 261 *A. fumigatus* genome assemblies from NCBI. Graph represents the proportion of genomes in NCBI containing the various permutations of the translocation breakpoints identified in A. No match: neither left nor right breakpoint flank found in target genome. R only: only right breakpoint flank found in target genome. L only: only left breakpoint flank found in target genome. Split BP: both right and left breakpoint flanks found in target genome at separate locations. Intact BP: breakpoint and flanking regions found in the target genome. Chr – chromosome.

translocation regions where > 100 kb regions had translocated. 400 bp regions representing 200 bp upstream and downstream flanking the breakpoint for both prototypical (Af293) and translocation (CEA10) breakpoint sites were chosen and are shown in Supplementary Data 5 and graphically in Fig. 6a. Genome assembly contigs were formatted for BLAST and searched with BLASTN using the sequences in Supplementary Data 4 as query and a tabular output. Outputs were assessed for presence of contiguous query, presence of query upstream or downstream at different locations, presence of only one upstream or downstream sequence or absence of any upstream or downstream sequence and results are shown in Fig. 6b.

### Reporting summary
Further information on research design is available in the Nature Research Reporting Summary linked to this article.

### Data availability
The.fasta sequence and.gff files of both A1160 and CEA10 strains generated in this study have been deposited in the National Library of Medicine (https://www.ncbi.nlm.nih.gov/) database under the accession numbers SAMN28487500 for A1160 and SAMN28487501 for CEA10 (Bioproject no PRJNA838920). Data are also available from the corresponding authors upon request. Transcript data from NCBI SRA (https://www.ncbi.nlm.nih.gov/sra/; Supplementary Data 3) archive

was used here to guide annotation and to generate a list of potential transcribed regions. Please refer to for details. 261 *A. fumigatus* genome assemblies available in NCBI (https://www.ncbi.nlm.nih.gov/assembly; Supplementary Data 4) were used to learn about conservation of translocation breakpoints in other genomes in the species.

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

## Acknowledgements

This study was funded by the Fungal Infection Trust (https://fungalinfectiontrust.org/) awarded to M.G.F. M.G.F. was also supported by the Wellcome Trust grant 208396/Z/17/Z awarded to P.B. and D.D. P.B. was supported by the NIHR Manchester Biomedical Research Centre. The authors would like to thank Michael Bromley for sourcing the CEA10 strain and allowing access to the Illumina reads.

## Author contributions

M.G.F. and P.B. conceived the idea and designed the study. M.G.F. received fundings for the study. M.G.F. and A.C. performed the experiments. M.G.F., A.C and P.B. assembled and analysed the initial data. PB supervised and interpreted the results. P.B. and M.G.F. wrote the original draft. D.D. revised and contributed to the final version of the manuscript. All authors agreed on the final content.

## Competing interests

The authors declare no competing interests.
