## [Peer Review File · Nature Communications]

Reviewers' Comments:

Reviewer #1:

Remarks to the Author:

This work presented here is an important and worthy contribution to our knowledge of fungal genetics. The manuscript itself seems incomplete. The authors describe this as a 'model' organism, which to me suggests it has relevance beyond itself or some narrow niche, but the discussion did not explore this aspect. Again, the work appears to be appropriate and valuable, but the discussion should connect and contextualize the results.

Details about the assembly methods are unclear or incomplete. It appears that for each strain there were 2 complete sequencing and assembly paths, but I can't see how the PacBio and Nanopore data / assemblies were brought together. Finally, almost as an afterthought at line 209, we see that Illumina data was also included.

At line 233 we see that data from NCBI SRA was used 'to guide annotation'. Can you explain which data were used? Was it *A. fumigatus* RNASeq data or something broader (other *Aspergillus* spp) or strain-specific data that already had been deposited?

I'm puzzled that there is no indication of deposit of sequence reads or assemblies into some public-accessible database beyond the supplemental data here.

I'm surprised that there such an abbreviated analysis of the data and limited discussion. I expected to see some analysis of the genome in the context of other *Aspergillus* spp., several of which have high-quality genomes available. What is the degree of synteny among the species? Conservation of physical arrangement of gene clusters?

Reviewer #2:

Remarks to the Author:

I have reviewed the manuscript titled "Telomere to telomere sequence of model *Aspergillus fumigatus* genomes by Bowyer et al. which is submitted for consideration to be published in Nature Communications. In this manuscript the authors report the sequencing and assembly of the genomes of two reference strains of *A. fumigatus*, CEA10 and A1160. In the study, sequencing reads were produced using the long-read sequencing technologies of PacBio Single Molecule Real-Time (SMRT) and Oxford Nanopore sequencing platforms. These long sequence reads were supplemented by paired-end Illumina reads to provide additional sequence coverage of the genome of CEA10. Genome assembly and annotations were accomplished providing complete high-quality telomere to telomere sequences of the fungal chromosomes except for a telomere of chromosome 6 of strain A1160 which was reported in the manuscript to not be completely assembled due to chromosome rearrangements. The number of ribosomal repeat units is also unresolved in these genome sequences. The first *A. fumigatus* reference genome sequences of 2005 (Af293) and 2008 (Af1163) lacked coverage of centromeres, the accurate representation of the ribosomal repeats, and a comprehensive presentation of chromosomal rearrangements due to the limitations of the technology of the time. These additional very high-quality genome sequences thus provide important reference resources for supporting the continuing studies of this pathogen that employ these two strains. These strains have become standard in such laboratory experiments due to their robust growth, ease in genome manipulation, and pathogenicity characteristics.

The sequencing technologies and informatic tools and approaches for genome assembly, annotation, and analysis are sound and state-of-the art.

A few comments and suggestions are provided for consideration by the authors.

1. In the manuscript, the collective terms for the sequencing technology employed in the study include "current long-read sequencing technology" (line 53), "third-generation sequencing technologies" (line 55), and "long read next generation sequencing" (line 194). Perhaps a single version for this collective term would be more appropriate.

2. In line 59, the genus name of *A. awamori* should be spelled out as it is the first mention of this fungus in the manuscript.
3. In lines 79-80 the statement is made that "...centromeres...are composed of transposons." In line 138-139 the manuscript states "that the centromeres of *A. fumigatus* chromosomes are enriched with transposable elements." The second statement seems likely to be the most accurate.
4. Line 94 uses the acronym "SNP" which perhaps should be defined as is done for "INDELS" in line 96.
5. In line 108 the word "species" should be replaced by "strains" as both sequenced strains are *A. fumigatus*.
6. In line 119 the "o" should be replaced by "of".
7. Line 191 - 192 reports how the purified DNA quality was assessed prior to running the sequencing without providing an indication of the actual DNA quality. Perhaps revising this sentence to something like "...and the quality of the DNA was assessed to be within the quality specifications required by PacBio and Oxford Nanopore on a 1% agarose gel, as well as ..."
8. Lines 223 and 224 describe how the Illumina reads were used to give the final CEA10 sequence. No statement is provided for how the final sequence of A1160 was determined.
9. The legend for Fig. 2 seems inadequate for understanding the features illustrated in the figure. The figure legend identifies the colored boxes representing three kinds of features represented by colored boxes of red, blue and light green. However in the figure, 10 rows of bars suggest that the figure includes up to 10 rows of features with only three categories identified. The text does indicate the mapping of RNAseq data to the genome, but no color is specified for where this is included in the figure. There is also a large red box that may be a presumptive centromere as implied in the legend. A more detailed description of the features in the figure may assist the readers in interpreting the figure.

Answers to the Reviewers

We wish to thank the Reviewers for assessing our work and for the constructive comments. Please see below our answers to the points raised. All the corrections are highlighted in yellow in the manuscript.

Reviewer #1 (R1):

1. This work presented here is an important and worthy contribution to our knowledge of fungal genetics. The manuscript itself seems incomplete. The authors describe this as a ‘model’ organism, which to me suggests it has relevance beyond itself or some narrow niche, but the discussion did not explore this aspect.

A1. The **R1** makes a good point since *A. fumigatus* is not model organism *per se* like for example *Saccharomyces cerevisiae*. However, *A. fumigatus* is arguably the model human mould pathogen. In fact, CEA10 is a progenitor strain for the most commonly used lineages of lab strain such as A1160 (Bertuzzi *et al.*, 2021), and it has been used for the whole genome knockout project and many protocols have been developed using it.

We have now changed the title to “Telomere to telomere sequence of the model mould pathogen *Aspergillus fumigatus* genomes” and clarified that in the lines 35-36.

Reference:

Bertuzzi M, van Rhijn N, Krappmann S, Bowyer P, Bromley MJ, Bignell EM. On the lineage of *Aspergillus fumigatus* isolates in common laboratory use. *Med Mycol* **59**, 7-13 (2021)

2. Again, the work appears to be appropriate and valuable, but the discussion should connect and contextualize the results.

Q2. We have now expanded and better contextualize the results and discussion on our data. Specifically, we **i.** clarified the sequencing and assembly workflow in the text and added a pipeline figure (Supplementary Fig. 1) and **ii.** performed an extensive analysis on conservation of translocation breakpoints in other species (lines 167-193, Fig. 5 and Supplementary Datasets 8 and 9).

3. Details about the assembly methods are unclear or incomplete. It appears that for each strain there were 2 complete sequencing and assembly paths, but I can’t see how the PacBio and Nanopore data / assemblies were brought together. Finally, almost as an afterthought at line 209, we see that Illumina data was also included.

A3. Yes, both strains were initially sequenced with PacBio and Oxford Nanopore technologies. The genome sequencing provided ~200x and ~40x coverage, for PacBio and Nanopore, respectively (see Supplementary Dataset 1 presented in the original paper for the metrics assembly). Since we already had in-house Illumina HiSeq sequences for the strain CEA10, we

used them for further validation of the final sequences for this strain. This is now clarified in the abstract – lines 23 and in the main text - lines 67-73, 85-87, 103-106 and 254-255. Additionally, a new Supplementary Fig. 1 with the pipeline used for the assembly of the genomes (lines 258-259) was added.

4. At line 233 we see that data from NCBI SRA was used ‘to guide annotation’. Can you explain which data were used? Was it *A. fumigatus* RNASeq data or something broader (other *Aspergillus* spp) or strain-specific data that already had been deposited?

A4. *A. fumigatus* RNAseq data was used to confirm genes and features after mapping protein and gene sequences to the genome with Exonerate and Augustus. We have now provided a list of all used NCBI SRA accession numbers in a new ‘Supplementary Dataset 7’ (line 282).

5. I’m puzzled that there is no indication of deposit of sequence reads or assemblies into some public-accessible database beyond the supplemental data here.

A5. The complete sequences generated in this study is now deposited in <https://www.ncbi.nlm.nih.gov/> with the accession numbers SAMN28487500 for A1160 and SAMN28487501 for CEA10 (lines 308-311).

6. I’m surprised that there such an abbreviated analysis of the data and limited discussion. I expected to see some analysis of the genome in the context of other *Aspergillus* spp., several of which have high-quality genomes available. What is the degree of synteny among the species? Conservation of physical arrangement of gene clusters?

A6. The synteny and comparison between *Aspergillus fumigatus* strains have been previously performed quite extensively using Af293 and other partial *A. fumigatus* assemblies (*i.e.* cluster arrangement and occurrence have also been already published; see references).

Following the **R1** advise, we have now also compared our data to Af293 and all *A. fumigatus* genome assemblies available in NCBI (total of 261) and added a section “Conservation of translocation breakpoints in other genomes in the species” (lines 167-193), a new Fig. 5 and all corresponding data in new Supplementary Dataset 8 and 9.

References:

Fedorova ND et al., Genomic islands in the pathogenic filamentous fungus *Aspergillus fumigatus*. *PLoS Genet.* 2008 Apr 11;4(4):e1000046. doi: 10.1371/journal.pgen.1000046

Kjærboelling I, Vesth TC, Frisvad JC, et al. Linking secondary metabolites to gene clusters through genome sequencing of six diverse *Aspergillus* species. *Proc Natl Acad Sci U S A.* 2018;115(4):E753-E761. doi:10.1073/pnas.1715954115

Lind AL, Wisecaver JH, Lameiras C, et al. Drivers of genetic diversity in secondary metabolic gene clusters within a fungal species. *PLoS Biol.* 2017;15(11):e2003583. Published 2017 Nov 17. doi:10.1371/journal.pbio.2003583

Steenwyk JL, Mead ME, Knowles SL, et al. Variation Among Biosynthetic Gene Clusters, Secondary Metabolite Profiles, and Cards of Virulence Across *Aspergillus* Species. *Genetics*. 2020;216(2):481-497. doi:10.1534/genetics.120.303549

Reviewer #2 (R2)

1. In the manuscript, the collective terms for the sequencing technology employed in the study include “current long-read sequencing technology” (line 53), “third-generation sequencing technologies” (line 55), and “long read next generation sequencing” (line 194). Perhaps a single version for this collective term would be more appropriate.

A1. The sequence technology has now been consistently referred to as “long read next generation sequencing” through-out the text.

2. In line 59, the genus name of *A. awamori* should be spelled out as it is the first mention of this fungus in the manuscript.

A2. This was corrected to: *Aspergillus awamori*.

3. In lines 79-80 the statement is made that “...centromeres...are composed of transposons.” In line 138-139 the manuscript states “that the centromeres of *A. fumigatus* chromosomes are enriched with transposable elements.” The second statement seems likely to be the most accurate.

A3. As suggested by **R2**, the sentence:

“Interestingly, we found that the centromeres of *A. fumigatus* encompass long stretches of DNA and are **composed of** transposons.”

is now replaced with:

“Interestingly, we found that the centromeres of *A. fumigatus* encompass long stretches of DNA and are **enriched with** transposons.”

Lines 91-93

4. Line 94 uses the acronym “SNP” which perhaps should be defined as is done for “INDELS” in line 96.

A4. All the acronyms are now fully explained.

5. In line 108 the word “species” should be replaced by “strains” as both sequenced strains are *A. fumigatus*.

A5. Corrected (line 124)

6. In line 119 the “o” should be replaced by “of”.

A6. Corrected (line 136)

7. Line 191 – 192 reports how the purified DNA quality was assessed prior to running the sequencing without providing an indication of the actual DNA quality. Perhaps revising this sentence to something like “...and the quality of the DNA was assessed to be within the quality specifications required by PacBio and Oxford Nanopore on a 1% agarose gel, as well as ...”

A7. The sentence:

“The DNA was eluted in 100 μ L of dH₂O and the quality of the DNA was assessed on a 1% agarose gel, as well as using a nanodrop (ThermoFisher Scientific) and a Qubit 4 Fluorometer (ThermoFisher Scientific)”

is now changed to:

“The DNA was eluted in 100 μ L of dH₂O and the quality of the DNA was assessed on a 1% agarose gel, as well as using a nanodrop (ThermoFisher Scientific) and a Qubit 4 Fluorometer (ThermoFisher Scientific) **to be within quality specification range required by the PacBio and Nanopore protocols.**”

Lines 236-237

8. Lines 223 and 224 describe how the Illumina reads were used to give the final CEA10 sequence. No statement is provided for how the final sequence of A1160 was determined.

A8. We did not use Illumina reads for A1160. This is now clarified in the abstract – lines 23 and in the main text - lines 67-73, 85-87, 103-106 and 254-255.

9. The legend for Fig. 2 seems inadequate for understanding the features illustrated in the figure. The figure legend identifies the colored boxes representing three kinds of features represented by colored boxes of red, blue and light green. However in the figure, 10 rows of bars suggest that the figure includes up to 10 rows of features with only three categories identified. The text does indicate the mapping of RNAseq data to the genome, but no color is specified for where this is included in the figure. There is also a large red box that may be a presumptive centromere as implied in the legend. A more detailed description of the features in the figure may assist the readers in interpreting the figure.

A9. Fig. 2 and its legend are now corrected as suggested by the R2.

Specifically: *i.* all features (rows) in the figure are now clearly labelled; *ii.* similar to Fig. 1, the centromere in Fig. 2 is now marked clearer with a green rectangle. In fact, for clearly we also included a color-coded legend to Fig. 1.

Reviewers' Comments:

Reviewer #1:

Remarks to the Author:

Looks good!

Minor point: Check at line 218: Should be "300uL mark"?

I really appreciate the pipeline schematic in Supplemental Figure 1.

Reviewer #2:

Remarks to the Author:

I agree with the comments of reviewer 1 and from my perspective the responses and manuscript revisions and additions in response to reviewer 1 comments are thorough and appropriate.

The responses and manuscript revisions and additions in response to reviewer 2 comments are thorough and appropriate.

Answers to the Reviewers

We wish to thank the Reviewers for assessing our work and for the constructive comments. Please see below our answers to the points raised. All the corrections are highlighted in yellow in the manuscript.

Reviewer #1:

Looks good!

Minor point: Check at line 218: Should be "300uL mark"?

Thank you for spotting this. The error was corrected.

I really appreciate the pipeline schematic in Supplemental Figure 1.

Reviewer #2:

I agree with the comments of reviewer 1 and from my perspective the responses and manuscript revisions and additions in response to reviewer 1 comments are thorough and appropriate.

The responses and manuscript revisions and additions in response to reviewer 2 comments are thorough and appropriate.